Pathways from marine protected area design and management to ecological success

Rudd Murray A. murray.a.rudd@emory.edu
Department of Environmental Sciences, Emory University , Atlanta, GA , United States
Esteban María Ángeles
Electronic publication date: 2015 Nov 26
Publication date: 2015
Volume: 3
Electronic Location ID: e1424
Received 2015 Jul 20; Accepted 2015 Nov 2
Copyright: © 2015 Rudd
Copyright year: 2015
Copyright holder: Rudd
License: This is an open access article distributed under the terms of the Creative Commons Attribution License, which permits unrestricted use, distribution, reproduction and adaptation in any medium and for any purpose provided that it is properly attributed. For attribution, the original author(s), title, publication source (PeerJ) and either DOI or URL of the article must be cited.
License URL: https://creativecommons.org/licenses/by/4.0/

Keywords: Marine reserves, Governance, Performance, Metrics, Marine conservation, Enforcement, MPA performance

Funding: The author received no funding for this work.

==============================
Using an international dataset compiled from 121 sites in 87 marine protected areas (MPAs) globally (Edgar et al., 2014), I assessed how various configurations of design and management conditions affected MPA ecological performance, measured in terms of fish species richness and biomass. The set-theoretic approach used Boolean algebra to identify pathways that combined up to five ‘NEOLI’ (No-take, Enforced, Old, Large, Isolated) conditions and that were sufficient for achieving positive, and negative, ecological outcomes. Ecological isolation was overwhelming the most important condition affecting ecological outcomes but Old and Large were also conditions important for achieving high levels of biomass among large fishes (jacks, groupers, sharks). Solution coverage was uniformly low (<0.35) for all models of positive ecological performance suggesting the presence of numerous other conditions and pathways to ecological success that did not involve the NEOLI conditions. Solution coverage was higher (>0.50) for negative results (i.e., the absence of high biomass) among the large commercially-exploited fishes, implying asymmetries in how MPAs may rebuild populations on the one hand and, on the other, protect against further decline. The results revealed complex interactions involving MPA design, implementation, and management conditions that affect MPA ecological performance. In general terms, the presence of no-take regulations and effective enforcement were insufficient to ensure MPA effectiveness on their own. Given the central role of ecological isolation in securing ecological benefits from MPAs, site selection in the design phase appears critical for success.

Introduction

In the face of multiple pressures on marine ecosystems and resources, the creation of marine protected areas (MPAs) has been advanced as a robust management approach for conserving aquatic ecosystems and habitats, and maintaining ecological resilience (Allison et al., 2003; Lubchenco et al., 2003; Roberts, 1997). MPAs may help maintain ecological connectivity, protect critical habitat, provide a refuge for commercial and threatened species, and increase the viability of adjacent fisheries over the long-term (Gell & Roberts, 2003; Halpern & Warner, 2002; Lester et al., 2009; Sumaila et al., 2000; Weigel et al., 2014). There has been increasing recognition and appreciation of the potential importance of the ecological, social, and political context within which MPAs are designed and implemented (Crawford et al., 2006; Huijbers et al., 2015; Pollnac, Crawford & Gorospe, 2001; Rudd et al., 2003; Soykan & Lewison, 2015; Vandeperre et al., 2011; Warner & Pomeroy, 2012). Even after two decades of intensive ecology and modeling (Lester et al., 2009; White et al., 2011), however, understanding the role that MPAs play in ameliorating multiple stressors and in the provision of benefits to humans remains an important international research priority (Parsons et al., 2014; Rudd, 2014).

Given broad and potentially conflicting goals for MPAs (Agardy et al., 2003; Brown et al., 2001; Jones, 2002) and the range and complexity of factors interacting to affect MPA performance (e.g., Edgar et al., 2014; Guidetti & Sala, 2007; Soykan & Lewison, 2015), it seems highly probable that multiple context-dependent pathways to ‘success’ exist. Empirical MPA studies typically focus on short-term ecological outcomes at limited scales, while MPA models are typically more abstract, focusing on ecological responses arising from MPAs over larger spatial and temporal scales (White et al., 2011). As Halpern (2014: 167) noted, however, while it may seem “we know a lot about what leads to MPA success or failure…the simultaneous assessment of how various factors affect MPA success has been missing ….” This is especially the case when the design, governance, and management attributes of MPAs are considered in conjunction with ecological factors.

Statistical analysis of the causal relationships between MPA design, management, and outcomes can be problematic when limited number of case studies are available, making it difficult to identify pathways from MPA design and management to ecological outcomes. Developments over the last 20 years in set-theoretic approaches for comparative case analysis now, however, offer an approach with which to analyze contextual complexity in small- and medium-n comparative studies. This configuration-oriented approach, commonly referred to as qualitative comparative analysis (QCA), explores connections between causally relevant conditions and outcomes using set theory (Goertz & Mahoney, 2012; Ragin, 1987; Schneider & Wagemann, 2012). Cases are defined in terms of sets, combinations of conditions and outcomes, and Boolean algebra is used to simplify logical statements describing how those combinations are related to relevant outcomes. Set-theoretic methodologies have become increasingly popular in the social sciences for assessing contextual complexity (Rihoux, 2013; Rihoux & Marx, 2013; Schneider & Wagemann, 2012) but their use has been relatively limited in fisheries and marine conservation research (but see Bodin & Österblom, 2013; Kosamu, 2015; Stokke, 2007; Sutton & Rudd, 2015).

In their global MPA analysis, Edgar et al. (2014) aggregated some 171,000 underwater abundance counts from Reef Life Survey scuba transect data collected from 964 sites in 87 international MPAs, and combined them in 121 international MPA/ecoregion groupings. Their analytical focus was on the influence of NEOLI (No-take, Enforced, Old, Large, Isolated) conditions on fish biomass and fish species richness. Their statistical analysis Edgar et al. (2014: 216) suggested that the conservation benefits of MPAs “increase exponentially with the accumulation of the five key features: no take, well enforced, old (>10 years), large (>100 km2), and isolated by deep water or sand” (as one reviewer pointed out, however, the exponential pattern was in back-transformed log response ratios of inside versus outside biomass, so results could be amplified). That dataset provides an opportunity to use a set-theoretic approach to test for context-dependent pathways from MPA design and management conditions to positive (and negative) ecological performance. My research questions were: (1) what combinations of NEOLI features interact to affect ecological performance metrics in MPAs? and (2) how do pathways to positive and negative MPA outcomes vary for different ecological performance metrics?

Methods

Data

The global MPA dataset contained information from 964 sites in 87 MPAs, which was aggregated into 121 MPA/ecoregion groupings (hereafter referred to simply as MPAs for simplicity) for analysis. Edgar & Stuart-Smith (2009) provide details on Reef Life survey methodology and Edgar et al. (2014) provide additional information about global survey procedures and data compilation. Their dataset is based on transects performed by trained volunteer scuba divers and represents in excess of 171,000 underwater abundance counts at 1,986 dive sites (Edgar et al., 2014).

Data analysis

Qualitative comparative analysis

QCA uses set theory relationships to identify configurations of conditions that, when present, are necessary or sufficient to lead to outcomes of interest (Ragin, 1987; Ragin, 2000; Schneider & Wagemann, 2012). Each case (i.e., one of 121 MPAs in this analysis) is considered as a configuration of causally relevant conditions (i.e., combinations of the presence or absence of NEOLI conditions) and an outcome (i.e., metrics of fish biomass or species richness). QCA comparatively identifies similarities and differences across cases where different, context-dependent paths lead to particular outcomes of interest (Rihoux, 2013). Boolean minimization algorithms in QCA software (Ragin & Davey, 2014) succinctly express causal regularities in the data. Results from analyses are contextual in that the causal power of a condition often depends on the presence of absence of other causal conditions.

To illustrate how QCA can provide information regarding pathways to successful ecological outcomes, consider how contextual social and governance factors influence ecological success in small-scale fisheries in Southeast Asia (Sutton & Rudd, 2015). Among 50 case studies, multiple pathways involving various combinations of social and governance conditions led to local ecological successes, defined on a scale from extremely degraded to thriving local fish stocks. One pathway to success involved the presence of a community organization involved in community-based fisheries management in combination with a degree of autonomy in local governance decision-making at the community level. A second pathway to positive ecological results arose when local fisheries were subsistence oriented, even in the absence of local decision-making powers. Another pathway involved strong local leadership: even with weak local decision-making powers and a market-oriented local fishery, positive ecological outcomes were observed, suggesting local leadership could help mobilize a degree of restraint in local fisheries that helped ameliorate some external stressors. Together these three alternative pathways accounted for 69% of cases where successful ecological outcomes were attained.

Data coding

Case conditions and outcomes may be coded as dichotomous ‘crisp sets’ that dichotomously classify variables as 0 or 1 (fully out of or in a set) or as ‘fuzzy sets’ that exhibit partial membership in the set of an ideal type (Ragin, 2000). Edgar et al. (2014) originally coded the NEOLI conditions into low-medium-high categories for each variable. As the most important differences in their study was between medium and high levels of NEOLI conditions (see Fig. 3, Edgar et al., 2014), and conditions in the middle of a scale provide no additional information useful for differentiating sets in QCA (membership of 50% in a condition’s set is the point of maximum ambiguity in QCA), I aggregated low and medium levels to form crisp set definitions of NEOLI conditions (Table 1). Each condition was thus either fully in or fully out as a member of each set. This aggregation was carried out based on patterns observed in Edgar et al.’s (2014) results and prior to any QCA data analysis.

Table 1 Number of MPAs belonging to NEOLI condition sets.

Condition	Edgar et al. coding	Crisp sets	Comments	
	Low	Med	High	Fully out	Fully in		
No-take	0	46	75	46	75	Referred to as governance or regulations by Edgar et al. (2014): low, openly fished; med, within MPA but with some fishing; high, no-take zone within MPA	
Enforced	12	27	82	39	82	Assessed by field survey teams (Edgar et al., 2014): low, ‘paper park’ with little control; med, moderate policing but with violations; high, well-enforced	
Old	19	38	64	57	64	Low, <5 yrs; Med, 5–10 yrs; High > 10 yrs	
Large	24	56	41	80	41	Low, <1 km2; Med, 1–100 km2; High > 100 km2	
Isolated	79	18	24	97	24	Low, shallow reef habitat continuous across MPA boundary; Med, small (1–20%) of zone boundary breached by shallow reef habitat; High > MPA zone isolated from fishing zone by depth or sand barriers	

All ecological outcomes in the Edgar et al. (2014) dataset were measured during reef scuba surveys as biomass or fish species richness per 250 m2. I log-transformed (ln[n +1]) them to fuzzy set membership values in a calibration process. If fish species richness or biomass exceeded the 90th percentile for that outcome across all 121 MPAs, they were considered fully in that condition’s set of successful outcomes; if fish species richness or biomass was less than the 10th percentile, they were considered fully out of the set (Table 2). The crossover was the point where an MPA with an overall fish biomass level of 14,765 g per 250 m2 would, for example, be assigned 0.50 membership in the set High biomass (and by implication 0.50 in a set NOT[High biomass]). There are no theoretical reasons for defining ‘high’ levels of outcomes at particular levels but some MPAs within the Reef Life Survey dataset were functionally pristine, so full set membership in a positive outcome (i.e., >90th percentile for that condition) should indicate performance that is truly high in the range of possibilities. Note that a biomass of 120,572 g per 250 m2 corresponds to approximately 4,800 kg per ha, a figure well in excess of values suggested as baselines to define pristine reef fish biomass (MacNeil et al., 2015).

Table 2 Summary of outcome set calibrations.

Outcome sets	Min	Mean	Max	Fuzzy membership calibration	ln(n + 1) (per 250 m2)	
				Fully out	Crossover	Fully in		
High biomass	4.07	9.78	12.38	7.5 (1,808)*	9.6 (14,765)	11.7 (120,572)	Total fish biomass (g)	
High large fish biomass	3.11	8.54	11.79	5.5 (245)	8.1 (3,295)	10.7 (44,356)	Total biomass (g) of large fish	
High damselfish biomass	0.00	6.46	11.05	3.5 (33)	6.4 (602)	9.3 (10,938)	Total biomass (g) of damselfish	
High grouper biomass	0.00	3.36	8.74	1.0 (3)	4.5 (90)	8.0 (2,981)	Total biomass (g) of groupers	
High jack biomass	0.00	3.89	10.93	3.0 (20)	6.3 (518)	9.5 (13,360)	Total biomass (g) of jacks	
High shark biomass	0.00	2.78	11.06	0.7 (2)	5.1 (164)	9.5 (13,360)	Total biomass (g) of sharks	
High fish species richness	0.77	2.79	4.12	1.5 (5)	2.7 (14)	3.8 (45)	All fish species	
High large fish species richness	0.04	1.27	2.39	0.2 (1)	1.1 (3)	1.9 (7)	Large fish (>300 mm) species	
Notes.

* Values in parentheses denote cut-off and cross-over biomass and species richness values prior to ln(n + 1) transformation.

MPAs with only one or two NEOLI conditions performed poorly and were statistically indistinguishable from fished sites (Edgar et al., 2014). The lower 10th percentile cut-off that defined outcomes as fully outside the set of successful outcomes should thus reflect truly poor levels of MPA performance. Table 3 details the coding for each MPA (starting with MPAs exhibiting all five NEOLI conditions at the top of the table, proceeding through groupings of MPAs with decreasing numbers of NEOLI conditions, and ending with those MPAs that had only a single NEOLI condition; when multiple sites from the 121 international MPA/ecoregion groupings are from a single large MPA, they are denoted, for example, as Galapagos a to Galapagos e).

Table 3 Case study condition and outcome coding.

		Membership in condition:	Level of membership in the set of:	
	Country	No-take regulations	Enforcement	Old age	Large area	Isolated location	[High biomass]	[High large fish biomass]	[High damselfish biomass]	[High grouper biomass]	[High jacks biomass]	[High shark biomass]	[High species richness]	[High large fish richness]	
15. Cocos National Park	Costa Rica	1	1	1	1	1	0.98	1.00	0.78	0.80	0.91	1.00	0.74	1.00	
34. Kermadec Marine Reserve	New Zealand	1	1	1	1	1	0.99	1.00	0.98	0.93	0.77	1.00	0.60	0.95	
37. Lord Howe Commonwealth MPA a	Australia	1	1	1	1	1	0.78	0.96	0.78	0.95	0.70	1.00	0.75	0.91	
40. Malpelo Flora and Fauna Sanctuary	Colombia	1	1	1	1	1	0.99	1.00	0.89	1.00	0.76	1.00	0.64	0.93	
20. Florida Keys National Marine Sanctuary b	United States	1	1	1	1	0	0.73	0.68	0.76	0.69	0.89	0.00	0.71	0.85	
72. Tsitsikamma National Park	South Africa	1	1	1	1	0	0.65	0.62	0.00	0.58	0.00	0.00	0.33	0.69	
52. Poor Knights Island Marine Reserve	New Zealand	1	1	1	0	1	0.65	0.71	1.00	0.00	0.79	0.00	0.39	0.73	
63. Shiprock Aquatic Reserve	Australia	1	1	1	0	1	0.88	0.88	0.52	0.73	0.86	0.00	0.90	1.00	
1. Aldinga Reef	Australia	1	1	1	0	0	0.40	0.63	0.23	0.00	0.00	0.00	0.44	0.56	
11. Cape Rodney to Okakari Point Marine Reserve	New Zealand	1	1	1	0	0	0.47	0.66	0.17	0.00	0.82	0.00	0.19	0.53	
12. Cathedral Cove Marine Reserve	New Zealand	1	1	1	0	0	0.64	0.83	0.31	0.00	0.91	0.00	0.26	0.60	
19. Florida Keys National Marine Sanctuary a	USA	1	1	1	0	0	0.91	0.89	0.82	1.00	0.42	0.00	0.85	1.00	
21. Fly Point-Halifax Park	Australia	1	1	1	0	0	0.63	0.77	0.41	0.64	0.49	0.68	0.72	0.80	
26. Governor Island Marine Nature Reserve	Australia	1	1	1	0	0	0.00	0.25	0.00	0.00	0.00	0.00	0.00	0.42	
29. Hanauma Bay Marine Life Conservation District	USA	1	1	1	0	0	0.52	0.46	0.43	0.00	0.36	0.00	0.76	0.77	
35. La Restinga-Mar de las Calmas MPA	Spain	1	1	1	0	0	0.57	0.71	0.59	0.87	0.47	0.00	0.42	0.99	
36. La Reserve Naturelle Marine de Cerb re Banyuls	France	1	1	1	0	0	0.38	0.49	0.58	0.50	0.00	0.00	0.39	0.44	
38. Lord Howe Island Marine Park a	Australia	1	1	1	0	0	0.80	0.96	0.93	0.55	0.69	0.87	0.69	0.65	
41. Maria Island Marine Reserve	Australia	1	1	1	0	0	0.29	0.51	0.00	0.00	0.29	0.40	0.25	0.58	
42. Marmion Marine Park a	Australia	1	1	1	0	0	0.32	0.38	0.63	0.00	0.00	0.00	0.53	0.70	
45. Mushi Mas Mingili Thila	Maldives	1	1	1	0	0	0.84	0.88	0.41	0.00	0.93	0.00	1.00	0.98	
47. Ningaloo Marine Park a	Australia	1	1	1	0	0	0.60	0.52	0.84	0.65	0.00	0.00	0.85	0.38	
48. Ningaloo Marine Park b	Australia	1	1	1	0	0	0.64	0.68	0.78	0.82	0.73	0.00	0.91	0.82	
54. Port Noarlunga Reef	Australia	1	1	1	0	0	0.47	0.69	0.52	0.00	0.00	0.00	0.57	1.00	
59. Rose Atoll National Wildlife Refuge	American Samoa	1	1	1	0	0	0.62	0.46	0.39	0.56	0.26	0.00	0.83	0.51	
66. Sund Rock Marine Preserve	USA	1	1	1	0	0	0.18	0.29	0.00	0.00	0.00	0.00	0.24	0.36	
73. Tuhua/Mayor Island marine reserve	New Zealand	1	1	1	0	0	0.66	0.79	0.89	0.00	0.85	0.71	0.39	0.88	
74. Tulamben	Indonesia	1	1	1	0	0	0.65	0.70	0.97	0.50	0.49	0.00	1.00	0.89	
27. Great Barrier Reef MP a	Australia	1	1	0	1	0	0.67	0.67	0.99	0.78	0.61	0.65	0.93	0.63	
33. Kent Group Marine Park a	Australia	1	1	0	1	0	0.59	0.51	0.53	0.00	0.00	0.50	0.37	0.66	
4. Beware Reef Marine Sanctuary	Australia	1	1	0	0	1	0.78	0.74	0.69	0.00	0.38	0.71	0.47	0.92	
70. Te Paepae o Aotea Marine Reserve	New Zealand	1	1	0	0	1	1.00	1.00	1.00	0.00	0.30	0.00	0.33	0.81	
3. Batemans Marine Park a	Australia	1	1	0	0	0	0.74	0.73	0.96	0.45	1.00	0.72	0.49	0.86	
5. Bushrangers Bay Aquatic Reserve	Australia	1	1	0	0	0	0.44	0.55	0.84	0.00	0.00	0.00	0.59	0.62	
6. Cabbage Tree Bay Aquatic Reserve	Australia	1	1	0	0	0	0.73	0.73	0.60	0.19	0.90	0.73	0.74	0.77	
9. Cape Byron Marine Park	Australia	1	1	0	0	0	0.88	0.98	0.84	0.60	0.64	0.86	0.86	1.00	
10. Cape Howe Marine National Park	Australia	1	1	0	0	0	0.78	0.68	0.58	0.00	1.00	0.78	0.50	0.93	
13. Channel Islands National Marine Sanctuary a	USA	1	1	0	0	0	0.33	0.58	0.00	0.00	0.00	0.00	0.30	0.65	
14. Channel Islands National Marine Sanctuary b	USA	1	1	0	0	0	0.58	0.67	0.98	0.00	0.00	0.11	0.43	0.81	
23. Galapagos Marine Reserve b	Ecuador	1	1	0	0	0	1.00	1.00	0.64	0.77	0.68	1.00	0.74	1.00	
28. Great Barrier Reef MP b	Australia	1	1	0	0	0	0.63	0.70	0.87	0.82	0.76	0.00	1.00	0.88	
31. Jervis Bay a	Australia	1	1	0	0	0	0.81	0.71	0.72	0.09	1.00	0.81	0.66	0.79	
32. Jurien Bay a	Australia	1	1	0	0	0	0.39	0.54	0.79	0.51	0.00	0.00	0.47	0.60	
50. Point Cooke Marine Sanctuary	Australia	1	1	0	0	0	0.00	0.00	0.00	0.00	0.00	0.00	0.00	0.00	
51. Point Lobos State Marine Reserve	USA	1	1	0	0	0	0.16	0.47	0.00	0.00	0.00	0.00	0.25	0.70	
53. Port Davey National Park a	Australia	1	1	0	0	0	0.00	0.07	0.00	0.00	0.00	0.42	0.00	0.17	
55. Port Phillip Heads Marine National Park	Australia	1	1	0	0	0	0.44	0.66	0.60	0.00	0.00	0.72	0.30	0.68	
56. Port Stephens Great Lake Marine Park a	Australia	1	1	0	0	0	0.71	0.57	0.69	0.25	1.00	0.68	0.67	0.70	
58. Rickett’s Point Marine Sanctuary	Australia	1	1	0	0	0	0.00	0.12	0.00	0.00	0.00	0.00	0.00	0.12	
60. Rottnest Island a	Australia	1	1	0	0	0	0.65	0.80	0.65	0.39	0.29	0.42	0.56	0.67	
65. Solitary Islands Marine Park a	Australia	1	1	0	0	0	0.88	1.00	0.94	0.74	0.88	1.00	0.69	0.70	
71. Tinderbox Marine Reserve	Australia	1	1	0	0	0	0.21	0.39	0.00	0.00	0.00	0.00	0.23	0.47	
17. Ponta da Baleia-Abrolhos a	Brazil	1	0	1	1	0	0.62	0.61	0.81	0.63	0.00	0.00	0.61	0.67	
25. Golfo de Chiriqui Marine National Park	Panama	1	0	1	1	0	0.33	0.13	0.72	0.85	0.44	0.00	0.57	0.17	
43. Mnazi Bay-Ruvuma Estuary Marine Park	Tanzania	1	0	1	1	0	0.59	0.34	0.78	0.67	0.00	0.00	1.00	0.70	
2. Baie Ternay	Seychelles	1	0	1	0	0	0.71	0.65	0.56	0.53	0.00	0.00	0.98	0.80	
30. Isla de Taboga e Isla de Uraba Wildlife Refuge	Panama	1	0	1	0	0	0.49	0.63	0.68	0.86	0.00	0.00	0.59	0.73	
39. Machalilla	Ecuador	1	0	1	0	0	0.45	0.52	0.79	0.84	0.00	0.00	0.64	0.67	
49. Pangaimotu Reef MPA	Tonga	1	0	1	0	0	0.38	0.40	0.77	0.94	0.00	0.00	0.90	0.12	
62. Sesoko Scientific Research Area	Japan	1	0	1	0	0	0.57	0.00	0.79	0.32	0.00	0.00	0.97	0.12	
64. Shiraiwazaki Marine Park	Japan	1	0	1	0	0	0.12	0.04	0.00	0.61	0.00	0.00	0.62	0.12	
67. Table Mountain National Park a	South Africa	1	0	1	0	0	0.59	0.53	0.00	0.00	0.03	0.45	0.23	0.52	
68. Tawharanui Marine Reserve	New Zealand	1	0	1	0	0	0.17	0.46	0.00	0.00	0.43	0.00	0.06	0.36	
75. Ushibuka Marine Park	Japan	1	0	1	0	0	0.40	0.24	0.57	0.75	0.00	0.00	0.66	0.29	
16. Coiba National Park a	Panama	1	0	0	1	1	0.76	0.85	0.80	0.91	0.85	0.57	0.69	0.71	
44. Motu Motiro Hiva	Chile	1	0	0	1	1	0.65	0.62	0.59	0.00	0.36	0.85	0.50	0.53	
7. Caletas	Costa Rica	1	0	0	1	0	0.13	0.00	0.45	0.66	0.00	0.00	0.42	0.00	
8. Camaronal	Costa Rica	1	0	0	1	0	0.06	0.12	0.56	0.62	0.00	0.00	0.39	0.00	
18. Fiordo Comau Protected Area	Chile	1	0	0	0	0	0.33	0.52	0.00	0.00	0.00	0.00	0.00	0.32	
22. Galapagos Marine Reserve a	Ecuador	1	0	0	0	0	1.00	1.00	0.98	1.00	0.77	1.00	0.74	1.00	
24. Galapagos Marine Reserve c	Ecuador	1	0	0	0	0	1.00	0.98	0.89	1.00	0.00	0.48	0.55	1.00	
46. Ninepin Point Marine Reserve	Australia	1	0	0	0	0	0.36	0.37	0.00	0.00	0.00	0.00	0.15	0.28	
57. Regno di Nettuno a	Italy	1	0	0	0	0	0.24	0.00	0.70	0.00	0.00	0.00	0.38	0.01	
61. Seaflower Area Marina Protegida a	Colombia	1	0	0	0	0	0.59	0.57	0.59	0.70	0.44	0.00	0.77	0.63	
69. Te Matuku Marine Reserve	New Zealand	1	0	0	0	0	0.00	0.00	0.00	0.00	0.00	0.00	0.00	0.00	
101. Lord Howe Commonwealth MPA b	Australia	0	1	1	1	1	0.75	0.85	0.82	1.00	0.69	0.88	0.75	0.88	
80. Channel Islands National Marine Sanctuary c	USA	0	1	1	1	0	0.29	0.44	0.35	0.00	0.00	0.00	0.44	0.69	
81. Channel Islands National Marine Sanctuary d	USA	0	1	1	1	0	0.56	0.42	1.00	0.00	0.00	0.00	0.41	0.57	
86. Florida Keys National Marine Sanctuary c	USA	0	1	1	1	0	0.61	0.55	0.73	0.62	0.40	0.00	0.76	0.59	
104. Ningaloo Marine Park c	Australia	0	1	1	1	0	0.74	0.55	0.80	0.77	0.00	0.70	0.94	0.70	
105. Ningaloo Marine Park d	Australia	0	1	1	1	0	0.69	0.76	0.82	0.68	0.03	0.77	0.88	0.63	
78. Bonaire	Netherlands Antilles	0	1	1	0	1	0.56	0.56	0.91	0.48	0.33	0.00	0.84	0.90	
87. Fly Point-Halifax Park	Australia	0	1	1	0	0	0.39	0.46	0.29	0.33	0.00	0.54	0.72	0.55	
102. Lord Howe Island Marine Park b	Australia	0	1	1	0	0	0.66	0.79	0.93	0.07	0.43	0.78	0.62	0.59	
103. Marmion Marine Park b	Australia	0	1	1	0	0	0.74	0.92	0.70	0.36	0.17	0.00	0.48	0.64	
106. North Sydney Harbour Aquatic Reserve	Australia	0	1	1	0	0	0.64	0.69	0.81	0.00	0.85	0.29	0.66	0.66	
114. Rottnest Island c	Australia	0	1	1	0	0	0.56	0.68	0.73	0.40	0.43	0.70	0.55	0.68	
116. Shoalwater Islands Marine Park	Australia	0	1	1	0	0	0.41	0.57	0.45	0.00	0.00	0.00	0.31	0.30	
118. St. Abbs and Eyemouth Marine Reserve	Scotland	0	1	1	0	0	0.05	0.31	0.00	0.00	0.00	0.00	0.07	0.27	
89. Galapagos Marine Reserve e	Ecuador	0	1	0	1	1	0.88	0.97	0.78	0.89	0.45	0.92	0.67	1.00	
112. Rose Atoll National Monument	American Samoa	0	1	0	1	1	0.66	0.58	0.49	0.47	0.70	0.00	0.85	0.71	
76. Batemans Marine Park b	Australia	0	1	0	1	0	0.74	0.77	0.90	0.18	0.99	0.78	0.49	0.85	
91. Great Barrier Reef MP c	Australia	0	1	0	1	0	0.64	0.63	0.92	0.58	0.26	0.64	0.91	0.47	
94. Jervis Bay b	Australia	0	1	0	1	0	0.78	0.73	0.94	0.00	1.00	0.88	0.60	0.72	
95. Jurien Bay b	Australia	0	1	0	1	0	0.43	0.65	0.71	0.40	0.41	0.00	0.43	0.63	
97. Kent Group Marine Park b	Australia	0	1	0	1	0	0.53	0.49	0.58	0.00	0.00	0.44	0.40	0.67	
100. Levante de Mallorca Cala Ratjada	Spain	0	1	0	1	0	0.19	0.26	0.37	0.00	0.00	0.00	0.50	0.12	
109. Port Stephens Great Lake Marine Park b	Australia	0	1	0	1	0	0.92	0.75	0.88	0.24	1.00	0.85	0.62	0.74	
117. Solitary Islands Marine Park b	Australia	0	1	0	1	0	0.70	0.74	0.87	0.19	0.56	0.76	0.55	0.49	
93. Illa del Toro	Spain	0	1	0	0	1	0.54	0.54	0.95	0.74	0.00	0.00	0.48	0.73	
79. Bronte-Coogee Aquatic Reserve	Australia	0	1	0	0	0	0.55	0.55	0.67	0.00	0.62	0.00	0.63	0.64	
92. Great Barrier Reef MP d	Australia	0	1	0	0	0	0.58	0.50	0.81	0.71	0.00	0.00	1.00	0.70	
108. Port Davey National Park b	Australia	0	1	0	0	0	0.01	0.22	0.00	0.00	0.04	0.37	0.00	0.32	
110. Pupukea Marine Life Conservation District	USA	0	1	0	0	0	0.30	0.21	0.30	0.00	0.00	0.00	0.63	0.36	
113. Rottnest Island b	Australia	0	1	0	0	0	0.32	0.39	0.71	0.62	0.00	0.00	0.58	0.62	
82. Coiba National Park b	Panama	0	0	1	1	1	0.85	1.00	0.71	0.94	1.00	0.80	0.65	0.73	
84. Coringa-Herald Nature Reserve	Australia	0	0	1	1	1	0.61	0.77	0.66	0.63	0.48	0.90	0.98	0.66	
85. Ponta da Baleia-Abrolhos b	Brazil	0	0	1	1	0	0.56	0.59	0.66	0.00	0.17	0.00	0.61	0.63	
119. Strangford Lough Marine Nature Reserve	N Ireland	0	0	1	1	0	0.00	0.00	0.00	0.00	0.00	0.00	0.00	0.00	
120. Table Mountain National Park b	South Africa	0	0	1	1	0	0.41	0.07	0.00	0.00	0.52	0.00	0.10	0.21	
77. Beacon Island Reef Observation Area	Australia	0	0	1	0	1	0.69	0.88	0.67	0.82	0.16	0.00	0.67	0.94	
83. Coral Patches Reef Observation Area	Australia	0	0	1	0	1	0.45	0.70	0.66	0.56	0.00	0.00	0.47	0.49	
99. Leo Island Reef Observation Area	Australia	0	0	1	0	1	0.65	0.72	0.72	0.59	0.00	0.00	0.60	0.77	
96. Kawasan Wisata	Indonesia	0	0	1	0	0	0.70	0.46	0.83	0.58	0.17	0.00	1.00	0.49	
107. Panglima Laut	Indonesia	0	0	1	0	0	0.62	0.30	0.79	0.56	0.00	0.00	1.00	0.21	
88. Galapagos Marine Reserve d	Ecuador	0	0	0	1	1	1.00	1.00	0.89	0.95	0.49	0.71	0.64	1.00	
90. Galapagos Marine Reserve f	Ecuador	0	0	0	1	1	0.94	1.00	0.88	1.00	0.00	0.43	0.57	1.00	
98. Las Perlas Marine Special Management Zone	Panama	0	0	0	1	1	0.84	0.80	0.93	0.97	0.73	0.49	0.72	0.81	
111. Regno di Nettuno b	Italy	0	0	0	1	0	0.15	0.00	0.67	0.00	0.00	0.00	0.41	0.01	
121. Wadi El Gemal—Hamata Reserve	Egypt	0	0	0	1	0	0.54	0.56	0.68	0.56	0.25	0.00	1.00	0.85	
115. Seaflower Area Marina Protegida b	Colombia	0	0	0	0	1	0.99	0.71	0.30	0.00	0.28	0.00	0.84	0.81	

Truth tables

Truth tables show the connection between all possible configurations of causal conditions that lead to an outcome of interest. The columns represent sets of causal conditions and an outcome, while the rows represent all logically possible intersections among the relevant sets. There is an exponential increase in configuration space as the number of conditions increases, with 2k ideal types for k conditions. Assignment of an MPA to a configuration was based on the MPA’s membership in each condition’s set and the number of empirical instances of each output of interest was recorded for each configuration. The default inclusion level was set at 0.70 (e.g., a configuration with inclusion = 0.72 would be deemed to ‘usually’ belong to the set High biomass) for testing sufficiency and 0.90 for testing necessity (see Rihoux & Ragin, 2009). In truth tables, outcomes with inclusion levels in excess of the cut-off were coded as successful (inclusion = 1); configurations not meeting the cut-off were coded as unsuccessful (inclusion = 0).

Necessary and sufficient conditions

A truth table forms a Boolean function that can be expressed as a union of fundamental set intersections, each of which corresponds to a successful outcome (Thiem & Duşa, 2013). If a condition is necessary for an outcome, the condition is a superset of the outcome (i.e., the outcome occurs only in the presence of the condition). If a condition is sufficient for an outcome, the condition is a subset of the outcome (i.e., the outcome always occurs only in the presence of the condition but also in its absence).

To give a practical example, suppose that high levels of fish biomass were observed in five MPAs, two of which allowed fishing and three of which prohibited fishing: prohibition of fishing would not be a necessary condition for the positive ecological outcome. However, if in the three cases where fishing was prohibited there were three positive outcomes and no negative outcomes, a prohibition on fishing could be sufficient for achieving the positive ecological outcome.

Boolean logic is used to simplify those set relations from the truth table to as few conditions as are defensible from a theoretical or empirical perspective. The level of Boolean minimization depends on assumptions made regarding the feasibility of the ‘logical remainders’, those configurations for which there are no empirical instances, and the minimum number of empirical instances needed for a configuration to be retained in a model (setting a cut-off level can help dampen noisiness arising from outlying cases). I used a default frequency cut-off of two empirical instances to assess necessary and sufficient conditions. If one assumes that, if observed, none of the logical remainders would result in a positive outcome, the result is the ‘complex solution’. On the other hand, if one assumes that all logical remainders would result in a positive outcome, a ‘parsimonious solution’ with the simplest possible sufficiency conditions results. These two solutions bound the complexity of the Boolean sufficiency conditions.

In QCA models, solution coverage assesses the extent to which a particular combination of causal conditions accounts for empirical instances of an outcome (Schneider & Wagemann, 2012). For example, overall coverage of 0.75 by two sufficient conditions would mean that of all empirical observations of the outcome of interest, 75% could be explained by one or the other (or both) of the conditions. Consistency, on the other hand, refers to the degree to which cases with a shared combination of causal conditions results in an outcome of interest (Schneider & Wagemann, 2012). For example, if a sufficient condition exhibited 0.80 consistency, 80% of all the occurrences of that particular combination of conditions would lead to the outcome of interest.

Models

Sixteen models were estimated in total, one each based on the presence or negation of each of the eight sets of ecological outcomes (species richness; species richness of large (>250 mm) fish; biomass of all fish; biomass of large fish; biomass of damselfish; biomass of jacks; biomass of groupers; biomass of sharks). I used Ragin’s fsQCA software (Ragin & Davey, 2014) for all analyses.

Results

With five NEOLI conditions, there were 32 possible combinations of conditions in each model. In total, 27 of the combinations had at least one empirical instance and 23 were observed at least twice and were retained for Boolean simplification (Table 4). Only four MPAs scored highly on all five NEOLI conditions; another five MPAs had various combinations of four of five possible NEOLI conditions (those nine were used by Edgar et al., 2014, as a baseline with which to compare non-fished and fished sites globally). Two configurations with four NEOLI conditions had no empirical instances and so were logical remainders in the QCA analysis.

Table 4 Number of empirical observations for each MPA configuration; configurations with no observations (logical remainders) and only one observation (below cut-off for inclusion in QCA model) are noted.

Configuration	No-take	Enforced	Old	Large	Isolated	Instances	Comments	
1	1	1	1	1	1	4	All 5 NEOLI conditions	
2	1	1	1	1	0	2	4 NEOLI conditions	
3	1	1	1	0	0	20	3 NEOLI conditions	
4	1	1	1	0	1	2	4 NEOLI conditions	
5	1	1	0	1	0	2	3 NEOLI conditions	
6	1	1	0	1	1	0	Logical remainder	
7	1	1	0	0	0	20	2 NEOLI conditions	
8	1	1	0	0	1	2	3 NEOLI conditions	
9	1	0	1	1	0	3	3 NEOLI conditions	
10	1	0	1	1	1	0	Logical remainder	
11	1	0	1	0	0	9	2 NEOLI conditions	
12	1	0	1	0	1	0	Logical remainder	
13	1	0	0	1	0	2	2 NEOLI conditions	
14	1	0	0	1	1	2	3 NEOLI conditions	
15	1	0	0	0	0	7	1 NEOLI condition	
16	1	0	0	0	1	0	Logical remainder	
17	0	1	1	1	0	5	3 NEOLI conditions	
18	0	1	1	1	1	1	Not included in analysis	
19	0	1	1	0	0	7	2 NEOLI conditions	
20	0	1	1	0	1	1	Not included in analysis	
21	0	1	0	1	0	8	2 NEOLI conditions	
22	0	1	0	1	1	2	3 NEOLI conditions	
23	0	1	0	0	0	5	1 NEOLI condition	
24	0	1	0	0	1	1	Not included in analysis	
25	0	0	1	1	0	3	2 NEOLI conditions	
26	0	0	1	1	1	2	3 NEOLI conditions	
27	0	0	1	0	1	3	2 NEOLI conditions	
28	0	0	1	0	0	2	1 NEOLI condition	
29	0	0	0	1	1	3	2 NEOLI conditions	
30	0	0	0	1	0	2	1 NEOLI condition	
31	0	0	0	0	1	1	Not included in analysis	
32	0	0	0	0	0	0	Logical remainder	

Necessary conditions

None of the five NEOLI conditions proved to be necessary for MPA ecological outcomes of interest (positive or negative) in any of the 16 models (Table 5). That is, in no case did a high level of ecological performance (or lack thereof) occur only in the presence of any single NEOLI condition. This is not unexpected given the complexity of potentially interacting factors influencing the ecological performance of MPAs.

Table 5 Tests of necessity for positive and negative ecological outcomes (must be ≥0.90 to be considered a necessary condition).

	Inclusion	Coverage	
High biomass			
No-take	0.61	0.55	
Enforced	0.70	0.57	
Old	0.53	0.56	
Large	0.38	0.62	
Isolated	0.28	0.79	
NOT[High biomass]			
No-take	0.63	0.45	
Enforced	0.65	0.43	
Old	0.53	0.44	
Large	0.29	0.38	
Isolated	0.10	0.21	
High large fish biomass			
No-take	0.62	0.58	
Enforced	0.73	0.63	
Old	0.54	0.59	
Large	0.35	0.61	
Isolated	0.28	0.83	
NOT[High large fish biomass]			
No-take	0.62	0.42	
Enforced	0.60	0.37	
Old	0.52	0.41	
Large	0.32	0.39	
Isolated	0.08	0.17	
High damselfish biomass			
No-take	0.58	0.55	
Enforced	0.69	0.61	
Old	0.52	0.58	
Large	0.40	0.69	
Isolated	0.26	0.77	
NOT[High damselfish biomass]			
No-take	0.68	0.45	
Enforced	0.66	0.39	
Old	0.55	0.42	
Large	0.26	0.31	
Isolated	0.11	0.23	
High grouper biomass			
No-take	0.62	0.39	
Enforced	0.57	0.33	
Old	0.60	0.44	
Large	0.45	0.52	
Isolated	0.33	0.64	
NOT[High grouper biomass]			
No-take	0.62	0.61	
Enforced	0.74	0.67	
Old	0.49	0.56	
Large	0.27	0.48	
Isolated	0.12	0.36	
High jack biomass			
No-take	0.66	0.35	
Enforced	0.81	0.39	
Old	0.51	0.31	
Large	0.41	0.40	
Isolated	0.30	0.50	
NOT[High jack biomass]			
No-take	0.60	0.65	
Enforced	0.61	0.61	
Old	0.54	0.69	
Large	0.30	0.60	
Isolated	0.15	0.50	
High shark biomass			
No-take	0.60	0.27	
Enforced	0.80	0.33	
Old	0.40	0.21	
Large	0.52	0.43	
Isolated	0.33	0.47	
NOT[High shark biomass]			
No-take	0.63	0.73	
Enforced	0.63	0.67	
Old	0.58	0.79	
Large	0.27	0.57	
Isolated	0.15	0.53	
High species richness			
No-take	0.59	0.54	
Enforced	0.67	0.56	
Old	0.57	0.60	
Large	0.37	0.61	
Isolated	0.23	0.66	
NOT[High species richness]			
No-take	0.66	0.46	
Enforced	0.69	0.44	
Old	0.48	0.40	
Large	0.30	0.39	
Isolated	0.16	0.34	
High large fish species richness			
No-take	0.63	0.62	
Enforced	0.74	0.68	
Old	0.53	0.62	
Large	0.34	0.63	
Isolated	0.27	0.83	
NOT[High large fish species richness]			
No-take	0.61	0.38	
Enforced	0.57	0.32	
Old	0.52	0.38	
Large	0.33	0.37	
Isolated	0.09	0.17	

Sufficient conditions

Model 1: High biomass

To illustrate QCA model interpretation, I present detailed results from the High biomass model before briefly summarizing the remaining models. Considering only configurations with two or more empirical instances each, 117 MPAs were represented in 23 NEOLI configurations. Of those, 17 cases in seven configurations exhibited inclusion levels >0.70 and were coded as members of (i.e., ‘usually in’) the set High biomass (Table 6). The High biomass set included the configuration where all five NEOLI conditions were present and one of the two observed configurations with four NEOLI conditions. The other configuration with four NEOLI conditions (No-take, Enforced, Old, Large) fell below the cut-off needed to ‘usually’ belong to the set High biomass.

Table 6 High biomass truth table: for MPAs with at least two observations, configuration counts and degree of membership inclusion in the set High biomass.

Configuration	No-take	Enforced	Old	Large	Isolated	Observed	High biomass	Inclusion	
1	1	1	1	1	1	4	1	0.935	
2	0	0	0	1	1	3	1	0.929	
3	1	1	0	0	1	2	1	0.890	
4	0	1	0	1	1	2	1	0.772	
5	1	1	1	0	1	2	1	0.764	
6	0	0	1	1	1	2	1	0.731	
7	1	0	0	1	1	2	1	0.703	
8	1	1	1	1	0	2	0	0.692	
9	0	0	1	0	0	2	0	0.663	
10	1	1	0	1	0	2	0	0.629	
11	0	1	0	1	0	8	0	0.617	
12	0	0	1	0	1	3	0	0.596	
13	0	1	1	1	0	5	0	0.578	
14	1	1	1	0	0	20	0	0.530	
15	1	1	0	0	0	20	0	0.518	
16	1	0	1	1	0	3	0	0.513	
17	1	0	0	0	0	7	0	0.501	
18	0	1	1	0	0	7	0	0.492	
19	1	0	1	0	0	9	0	0.430	
20	0	1	0	0	0	5	0	0.353	
21	0	0	0	1	0	2	0	0.344	
22	0	0	1	1	0	3	0	0.324	
23	1	0	0	1	0	2	0	0.098	

In the model’s complex solution (Eq. 1.C), five different pathways, derived by Boolean manipulation of the combinations of conditions in rows 1–7 of Table 6, were sufficient to result in high levels of fish biomass. Five conditions in each pathway are combined by logical AND operators; upper case denotes presence of condition and lower case denotes absence of a condition; dash denotes that a condition can be either present or absent; + denotes logical OR. To illustrate, the first condition,-eoLI, can be interpreted as follows: to achieve high levels of overall fish biomass via pathway 1.C1, the MPA can be either fished or not (i.e., N has no effect) AND enforcement is absent AND the MPA is not more than 10 years old AND the MPA is larger than 100 km2 AND the MPA is ecologically isolated (note that this corresponds to a total of 5 MPAs in the sample, the configuration represented by rows 2 and 7 in Table 6). In aggregate, the five pathways in the complex solution in combination provided 0.211 coverage and their level of aggregate inclusion was 0.838. The five pathways themselves each provided between 0.049 and 0.078 raw coverage individually; unique coverage for each pathway ranged from 0.021 to 0.055 and inclusion levels ranged from 0.827 to 0.878. ({1.C}) -eoLI+n-oLI+ne-LI+NE-lI+NEO-I→Highbiomass.

With the exception of Isolated, the conditions that formed pathways to High biomass could have either positive or negative effects on overall fish biomass depending on the context in which they occurred. While it may at superficially appear that Isolated is necessary for High biomass outcomes (because Isolated appears in each of the five pathways that combine to lead to High biomass), recall that the definition of a necessary condition is that it is a superset of the outcome: the outcome only appears in the presence of the condition. Table 6 showed, however, that row 12 (neO l I) had three empirical instances where High biomass was not achieved even though the MPAs were isolated; if Isolated were a necessary condition, these MPAs would also have exhibited a High biomass outcome. Neither can one say that Isolated is sufficient, on its own, to lead to positive outcomes. Considering only cases where High biomass was observed—the first seven rows of Table 6—Isolated appears in all configurations but never on its own, only in combination with other conditions in the seven distinct MPA configurations that lead to High biomass. Eight of nine logical remainders included Isolated, further highlighting the potential nuance of the role of ecological isolation—the complex solution assumes that none of the logical remainders would, if actually observed, result in high biomass (a point we return to in the Discussion).

Solution (1.C) exhibited configurational complexity in that multiple alternative pathways led to a single outcome of interest. The first three pathways comprising the complex solution highlighted that large, isolated MPAs could compensate, in terms of overall production of fish biomass, for some fishing within MPAs or in the face of weak enforcement, even in young MPAs. Pathways three and four implied that Large and the combination of No-take and Enforced were substitutes in the production of high levels of fish biomass within MPAs.

When the five pathways in the complex solution were simplified as much as possible with Boolean logic (i.e., assuming all logical remainders, if actually observed, would result in High biomass), the parsimonious solution (1.P) consisted of two pathways sufficient for achieving high levels of fish biomass within MPAs: ({1.P}) LI+EI→Highbiomass

Based on the ecological performance of the 17 MPAs that surpassed a reasonable threshold that qualified them as members of the set High biomass, ecological isolation in combination with either large area or effective enforcement were the simplest configurations that led to High biomass on a consistent basis. MPAs needed have only two NEOLI conditions and neither solution involved the presence of either No-take or Old. Moving from the complex to parsimonious solution increased coverage slightly from 0.211 to 0.238 and reduced inclusion from 0.838 to 0.804. The parsimonious pathways did not demonstrate the same level of subtlety as did the more stringent complex model. In the complex solution, a total of 17 specific MPAs were covered by the five sufficient pathways leading to High biomass (Table 7). A total of 20 MPAs were covered under the less stringent parsimonious solution.

Table 7 MPAs with greater than 50% membership in the sufficient condition High biomass.

Case	Complex	Parsimonious	
	1.C1	1.C2	1.C3	1.C4	1.C5	1.P1	1.P2	
4. Beware Reef Marine Sanctuary				1			1	
15. Cocos National Park					1	1	1	
16. Coiba National Park b	1					1		
34. Kermadec Marine Reserve					1	1	1	
37. Lord Howe Commonwealth MPA a					1	1	1	
40. Malpelo Flora and Fauna Sanctuary					1	1	1	
44. Motu Motiro Hiva	1					1		
52. Poor Knights Island Marine Reserve				1	1		1	
63. Shiprock Aquatic Reserve				1	1		1	
70. Te Paepae o Aotea Marine Reserve				1			1	
78. Bonaire							1	
82. Coiba National Park			1			1		
84. Coringa-Herald Nature Reserve			1			1		
88. Galapagos Marine Reserve d	1	1	1			1		
89. Galapagos Marine Reserve e		1				1	1	
90. Galapagos Marine Reserve f	1	1	1			1		
93. Illa del Toro							1	
98. Las Perlas Marine Special Management Zone	1	1	1			1		
101. Lord Howe Commonwealth MPA b						1	1	
112. Rose Atoll National Monument		1				1	1	
Notes.

Italics indicate MPAs not covered in the complex solution but covered under the parsimonious solution.

There were 14 cases with greater than 0.50 membership in pathway LI [Large AND Isolated], 13 cases with greater than 0.50 membership were covered by pathway EI [Enforced AND Isolated], and an overlap of 7 cases. Figure 1A shows an area-proportional Venn diagram (Micallef & Rodgers, 2014) with the set High biomass normalized to 100%. Solution coverage was 0.238 (0.065 + 0.089 + 0.084): the two solution pathways covered 23.8% of the area the set High biomass and the solution inclusion was 0.804 (i.e., 19.6% of the area of the two sufficient pathways fell outside of the High biomass set, in the set NOT[High biomass]). In Fig. 1B, the MPAs are mapped onto the sets of conditions and outcome. Low coverage left much of the set of High biomass unexplained and implies other conditions and sufficiency pathways are important in explaining high levels of overall fish biomass at the 121 MPAs.

Figure 1 Parsimonious solution for High biomass outcomes: (A) solution coverage by each of two pathways sufficient to achieve High biomass; and (B) specific MPAs that are members of pathways.

A more stringent consistency cut-off in the model would constrain solution boundaries. For example, increasing the inclusion cut-off to 0.85 in the first stage of the modeling process (i.e., imposing a more stringent definition of ‘usually in’ the set High biomass) would reduce the number of cases used in the Boolean analysis to nine empirical instances (coverage = 0.145) arising from three different configurations of MPA conditions (all still involving Isolated).

Model 2: NOT[High biomass]

In addition to analysis of positive ecological outcomes from MPAs, the QCA analysis in the second model identified sufficient conditions needed to lead to set negation, the set of all MPAs belonging to NOT[High biomass]. Two cases in a single configuration exhibited consistency levels >0.70 and were coded as members of the set NOT[High biomass]. The complex solution could not be simplified, coverage was 0.034, and a single pathway (Eq. 2.C/2.P) described a sufficient condition leading to the absence of high levels of fish biomass: ({2.C/2.P}) NeoLi→NOT[High biomass]

This pathway consisted of a very specific configuration involving all five conditions (2 present, 3 absent) and had only two empirical instances, the Costa Rican Caletas (row 7) and Camaronal (row 8) MPAs. NEOLI conditions played an extremely limited role in explaining pathways to low levels of overall fish biomass (Fig. 2). Some 97% of low biomass outcomes could not be explained by this solution, implying that other pathways not dependent on either the presence or absence of NEOLI conditions explained low levels of fish biomass (note the striking contrast in low biomass outcomes when comparing all fish species to commercially exploited species—see models 8, 10, and 12 below).

Figure 2 Venn diagram of solution coverage for model 2, NOT[High biomass].

Model 3: High large fish biomass

The complex solution (3.C) consisted of five pathways sufficient to lead to High large fish biomass. Table 8 shows model diagnostics for all parsimonious solutions leading to positive ecological outcomes. The parsimonious solution (3.P) consisted of a single pathway comprised of a single condition, ecological isolation. Three MPAs that were not part of the complex solution (78. Bonaire; 101. Lord Howe Commonwealth MPA b; 115. Seaflower Area Marina Protegida b) were part of the parsimonious solution simply by virtue of their ecological isolation. ({3.C}) neO-I+-eoLI+n-oLI+NE-lI+NEO-I→Highlargefishbiomass

({3.P}) —-I→Highlargefishbiomass

Table 8 Summary of parsimonious QCA solutions for positive outcomes (conditions, performance, and total coverage of MPAs by pathway and models).

Set/pathways to membership in set	MPA NEOLI conditions in sufficient solution	MPAs covered	Model performance	Pathway performance	
	No-take	Enforced	Old	Large	Isolated		Overall coverage	Overall inclusion	Raw coverage	Unique coverage	Pathway inclusion	
Model P.1: High biomass						20	0.238	0.804				
Pathway 1	–	–	–	L	I	14			0.173	0.084	0.835	
Pathway 2	–	E	–	–	I	13			0.154	0.065	0.803	
Model P.3: High large fish biomass						20	0.280	0.827				
Pathway 1	–	–	–	–	I	20			0.280	0.280	0.827	
Model P.5: High damselfish biomass						32	0.348	0.780				
Pathway 1	N	E	–	–	I	8			0.093	0.093	0.831	
Pathway 2	n	e	–	l	i	2			0.022	0.022	0.806	
Pathway 3	n	e	o	–	I	4			0.042	0.042	0.754	
Pathway 4	–	E	o	L	i	10			0.107	0.021	0.769	
Pathway 5	n	E	–	L	i	13			0.137	0.052	0.760	
Pathway 6	N	e	O	L	–	3			0.032	0.032	0.766	
Model P.7: High grouper biomass						13	0.240	0.870				
Pathway 1	–	–	O	L	I	7			0.132	0.099	0.892	
Pathway 2	n	e	–	L	I	5			0.095	0.062	0.897	
Pathway 3	N	e	O	L	–	3			0.046	0.046	0.716	
Model P.9: High jacks biomass						10	0.183	0.729				
Pathway 1	–	–	O	L	I	7			0.134	0.037	0.759	
Pathway 2	N	–	O	–	I	6			0.120	0.000	0.798	
Pathway 3	–	E	O	–	I	8			0.146	0.008	0.726	
Model P.11: High shark biomass						9	0.237	0.890				
Pathway 1	–	–	O	L	I	7			0.195	0.195	0.941	
Pathway 2	N	e	–	–	I	2			0.042	0.042	0.713	
Model P.13: High species richness						11	0.128	0.794				
Pathway 1	n	e	–	l	i	2			0.029	0.029	1.000	
Pathway 2	n	e	o	–	i	2			0.021	0.021	0.705	
Pathway 3	N	e	O	L	i	3			0.032	0.032	0.726	
Pathway 4	–	E	o	L	I	2			0.022	0.022	0.756	
Pathway 5	–	e	O	L	I	2			0.024	0.024	0.818	
Model P.15: High large species richness						21	0.240	0.854				
Pathway 1	–	–	–	l	I	10			0.108	0.087	0.808	
Pathway 2	n	–	o	–	I	7			0.081	0.060	0.867	
Pathway 3	N	E	O	L	–	6			0.071	0.071	0.887	
Notes.

Uppercase/bold denotes presence required; lowercase denotes absence required (i.e., NOT set member); dash denotes condition may be present or absent.

Model 4: NOT[High large fish biomass]

In total, seven MPAs were used to calculate a parsimonious solution (4.P) that included two sufficient pathways to membership in NOT[High large fish biomass] (Table 9 shows parsimonious solutions for all negated models; all complex solutions are available from the author upon request). In model 4, the parsimonious and complex solutions (4.C) coincided as no Boolean simplifications were possible. Note that, contrary to received wisdom about MPA size, Large figured in both sufficient pathways to low biomass outcomes; but recall it was also a condition in two pathways to High large fish biomass, demonstrating that the effect of MPA size was highly context dependent. ({4.P/4.C}) -eoLi+ne-Li→NOT[High large fish biomass]

Models 5–16

The remaining models for various ecological outcomes are outlined in Tables 8 and 9. Note that in Model 8 a total of 40 cases in seven configurations were coded as members of the negated set NOT[High grouper biomass]. This was the first model to achieve over 0.50 coverage, where four sufficient pathways in the parsimonious solution accounted for over half of all observed MPAs with low levels of grouper biomass. Similarly, 42 cases (coverage = 0.542) in Model 10 and 54 cases in Model 12 (coverage = 0.706) were coded as members of the negated sets NOT[High jack biomass] and NOT[High shark biomass], respectively. For all three commercially-targeted species, negated solutions had much higher coverage levels compared to more general biomass or species richness outcomes. Lack of ecological isolation was an important factor in virtually all solution pathways (Table 9).

Table 9 Summary of parsimonious QCA model solutions for negated models: conditions, performance, and total coverage of MPAs by pathway and models.

Set/pathways to membership in set	MPA NEOLI conditions	MPAs covered	Model performance	Pathway performance	
	No-take	Enforced	Old	Large	Isolated		Overall coverage	Overall inclusion	Raw coverage	Unique coverage	Pathway inclusion	
NEGATIVE OUTCOMES												
Model P.2: NOT[High biomass]						2	0.034	0.903				
Pathway 1	N	e	o	L	i	2			0.034	0.034	0.903	
Model P.4: NOT[High large fish biomass]						7	0.113	0.807				
Pathway 1	–	e	o	L	i	4			0.066	0.038	0.829	
Pathway 2	n	e	–	L	i	5			0.075	0.047	0.755	
Model P.6: NOT[High damselfish biomass]						3	0.048	0.781				
Pathway 1	n	e	O	L	i	3			0.048	0.048	0.781	
Model P.8: NOT[High grouper biomass]						40	0.519	0.782				
Pathway 1	n	–	o	–	i	15			0.156	0.087	0.768	
Pathway 2	–	E	o	l	–	20			0.286	0.233	0.755	
Pathway 3	n	E	–	l	–	14			0.139	0.086	0.735	
Pathway 4	n	e	–	L	i	5			0.060	0.041	0.887	
Model P.10: NOT[High jack biomass]						42	0.542	0.864				
Pathway 1	–	e	–	–	i	20			0.305	0.254	0.885	
Pathway 2	n	–	–	l	–	19			0.203	0.118	0.827	
Pathway 3	n	–	O	–	i	17			0.170	0.056	0.814	
Model P.12: NOT[High shark biomass]						54	0.706	0.905				
Pathway 1	–	e	–	–	i	20			0.299	0.132	0.931	
Pathway 2	n	–	o	l	–	7			0.076	0.076	0.947	
Pathway 3	–	–	O	L	i	13			0.132	0.040	0.887	
Pathway 4	–	–	O	l	I	6			0.069	0.069	1.000	
Pathway 5	N	–	O	–	i	20			0.354	0.199	0.909	
Model P.14: NOT[High species richness]						3	0.044	0.765				
Pathway 1	n	e	O	L	i	3			0.044	0.044	0.765	
Model P.16: NOT[High large species richness]						5	0.090	0.831				
Pathway 1	N	e	o	L	i	2			0.043	0.043	1.000	
Pathway 2	n	e	O	L	i	3			0.047	0.047	0.718	

Discussion

This study identified pathways that led from MPA design and management conditions to MPA performance, measured in terms of fish species richness and biomass for all fish, large fish, and specific groups of species (damselfish, groupers, jacks, sharks). The results demonstrated the importance of considering ecological and managerial conditions in the MPA design and implementation process. In addition to the substantive results, this study demonstrated the potential utility of QCA and set theory to assess the determinants of MPA performance and, more generally, how set theoretic approaches to ecological success may complement and extend insights from statistical analyses.

Conditions and configurations influencing ecological success

One of the five NEOLI conditions—ecological isolation—was pivotal for positive ecological outcomes relating to species richness of large fish and biomass of all fish, of large fish only, and of three commercially exploited fishes (groupers, jacks and sharks). Isolation was in fact, on its own, sufficient for leading to high biomass of large fish in the parsimonious solution. Isolated was present in 12 of 14 pathways to positive ecological performance in parsimonious solutions (Table 8) and NOT[Isolated] was not present in any. For complex solutions (available from the author upon request), Isolated was present in 20 of 22 pathways to positive ecological performance (NOT[Isolated] was present in only one solution). Conversely, in the negated models examining conditions influencing poor MPA performance, NOT[Isolated] was present in 14 of 17 pathways in parsimonious solutions (Table 9) (Isolated was present in one solution) and 17 of 21 in complex solutions.

Edgar et al. (2014) found that ecological isolation was important, seeming to “exert a stronger influence for community-level biomass and richness metrics than the other four features…(and that) although very important, the effect of isolation was similar in magnitude—rather than clearly superior—to other MPA features for biomass of sharks, groupers and jacks” (p. 218). QCA results instead suggest that ecological isolation is clearly the most important factor affecting ecological performance. The importance of isolation aligns well with insights from ecological models of MPAs (e.g., White et al., 2011).

Other areas of discrepancy between the statistical and set-theoretic models included the importance of: no-take regulations in the production of overall fish biomass (not part of QCA pathways to success in the parsimonious solutions and conflicting in direction in the complex solutions); MPA age being related to higher levels of jack biomass; and the effects of No-take and Enforced on species richness in MPAs (QCA results suggested that Old and Isolated also play important contextual roles). On other ecological outcomes, however, the approaches converged. For example, both statistical and set theoretic approaches identified lack of enforcement as being associated with relatively low levels of grouper biomass and identified the role of old MPAs in the production of high levels of shark biomass.

All conditions other than ecological isolation could positively or negatively affect positive ecological outcomes for large species diversity and biomass measures, thus highlighting the importance of context on success. Large MPAs appeared to, on balance, be important for positive outcomes and older MPAs appeared to be more important for commercially landed species compared to large species in general. MPA no-take regulations and enforcement did not have any degree of clear directional influence on ecological outcomes; in particular the NEO combination (row 14, Table 6), comprising 20 observations, had no positive or few minor negative impacts on ecological performance. Figure 3 illustrates all possible overlapping set combinations for the five NEOLI conditions and the number of empirical instances for each configuration. MPA performance, measured as the difference in the number of times a NEOLI condition was part of positive and negative parsimonious solutions, suggests that MPA ecological outcomes were broadly mediocre in the absence of ecological isolation.

Figure 3 Count of MPAs exhibiting various combinations of conditions.

Gray fill indicates configurations that were absent or observed at only a single site in the Reef Life Survey dataset. The colored configurations indicate the difference in times that particular configurations were present in parsimonious solutions less the times they appeared in negative solutions: blue fill indicates top performing MPA configurations (positive minus negative outcomes = +5, +6 or +7); green, +2, +3 or +4; yellow, −1, 0 or +1; orange, −4, −3 or −2; and red, −5, −6 or −7.

Overall, NEOLI conditions played a relatively limited role in sustaining high levels of ecological performance in MPAs. Solution coverage ranged from 0.128 for the species richness model to 0.348 for the damselfish biomass model, implying that 65% or more of positive ecological outcomes observed in the field could not be explained in terms of NEOLI conditions alone or in combination. On the other hand, the higher levels of coverage in the negated models of jack (0.542), grouper (0.519) and shark (0.706) biomass lends support for the perspective that MPAs may provide performance asymmetries and be more effective in preventing further declines in large fish biomass relative to rebuilding biomass towards levels seen in near-pristine conditions. Among the negated models, there were large differences in solution coverage between biomass outcomes for the large commercially-exploited species and the more general biomass and species richness models. This hints that there may be potential economic benefits for capture fisheries and tourism (for wildlife viewing) from conservation-oriented MPAs that provide insurance against declines in biomass of relatively mobile large species.

For damselfish biomass, biomass levels for all fish, and for large fish only, and for both species richness metrics, the negated models only covered 3%–11% of all negative outcomes. For the models with low coverage, the implication is that conditions other than the NEOLI conditions account for the vast majority of MPA buffering capacity against adverse ecological outcomes. A wide variety of other conditions have been identified as potentially important for MPA and small-scale fisheries management; some possible candidates include fishing community leadership, residents’ perceptions regarding threats to fish stocks, the availability of alternative livelihood opportunities, high levels of community engagement, management accountability, social capital and trust among community members, and outside (e.g., NGO) support for local management (e.g., Gutiérrez, Hilborn & Defeo, 2011; Pollnac, Crawford & Gorospe, 2001; Rudd et al., 2003; Warner & Pomeroy, 2012; Sutton & Rudd, 2015). Given the variability even among the eight indicators used in this study, it may also be the case that NEOLI conditions positively affect levels of alternative performance metrics. Many ecological indicators of MPA success are possible (Soykan & Lewison, 2015) and futher investigation would be needed to clarify the relationship between NEOLI conditions and a broader suite of ecological outcomes. The disparities in QCA coverage for different ecological metrics highlights the difficulties in relying on MPAs as robust tools for providing multiple types of conservation benefits simultaneously. MPAs may need to be explicitly focused on particular conservation goals rather than being implemented with unrealistic expectations that they can be ‘all things for all people’. Indeed, over a decade ago Agardy et al. (2003) cautioned that if MPAs failed to live up to unrealistically high expectations, there could be repercussions for marine conservation if managers were to lose confidence in MPAs as an effective tool in the overall conservation toolkit.

QCA utility for MPA studies

Set-theoretic methodologies have been used over the past 20 years to identify causal pathways from case conditions to outcomes of interest for a diverse range of social and political phenomena. Increasingly QCA has been applied in other fields such as the health sciences (Candy et al., 2013), public and social policy (Rihoux & Marx, 2013), and environmental management (Basurto, 2013; Huntjens et al., 2011; Never & Betz, 2014; Robinson, Holland & Naughton-Treves, 2014; Rudel, 2008; Sutton & Rudd, 2015). If used in conjunction with statistical and qualitative research, set-theoretic methods also have potential to help bridge the quantitative and qualitative research worlds (Brady & Collier, 2004; Goertz & Mahoney, 2012; Rihoux, 2003).

While beyond the scope of the current study, it would be possible to delve more deeply into context at MPAs that have been flagged as having potentially anomalous performance given their NEOLI configuration and develop hypotheses about which additional conditions, if empirically present, may or may not lead to outcomes of interest. Individual cases identified as logical remainders or contradictions in QCA can provide rich insights to help interpret Boolean solutions and can complement statistical analyses even in large-n studies (Glaesser & Cooper, 2011) by helping advance theoretical insights on determinants of MPA success. They can also provide direction for future sampling strategies—for example, it would seem to make sense for set theoretic analyses to have more sites that combined ecological isolation with various combinations of the other NEOLI conditions (recall Fig. 3) or to identify MPAs that, if they exist, did not have exhibit of the NEOLI conditions.

In addition to identifying potential case studies of research interest, QCA encourages substantive thinking about counterfactuals. In this study, for example, simply assuming that all remainders for ecologically isolated MPAs would result in High biomass (Eq. 1.P) is likely simplistic (given three empirical instances with the configuration neO l I did not result in High biomass). However, simply taking the complex model with its assumptions regarding a total lack of successful outcomes irrespective of conditions exhibited by the configurations with less than two empirical observations (Eq. 1.C) is somewhat naïve given two of the remainders involved combinations of four of the five NEOLI conditions. As all logical remainders and excluded cases lacked ecologically isolation (recall Fig. 3), it seems more plausible that parsimonious models are justifiable but this is an issue that could be explored in more depth (and may suggest strategic future sampling for additional Reef Life surveys).

Conclusions

There is a need for reflection on the design, governance, and management determinants of MPA success so that marine conservation investments can achieve the best possible ecological and socio-economic outcomes. Edgar et al. (2014) used the Reef Life Survey dataset to conduct a global analysis of MPA ecological performance and concluded that the conservation benefits of MPAs increased exponentially with the accumulation of the five NEOLI features. Halpern (2014), in the accompanying Nature editorial, argued that “It is clear that designating and enforcing park boundaries, although necessary, is not sufficient to gain full conservation benefits, and that protected areas without all five features should not be expected to produce such benefits” (p. 168). In this set theoretic analysis, results clearly showed that the presence of all five NEOLI were not needed to produce ecological benefits and that, when considering the five NEOLI conditions, only 20%–30% of cases with positive ecological outcomes were explained. The ‘bad news’ from this analysis was that set theoretic QCA does not support the view that accumulation of ecological benefits was exponential from three NEOLI conditions.

There is, however, a ‘good news’ story here as well. One of the NEOLI conditions—ecological isolation—was overwhelmingly the most important condition in the variety of configurations that led to positive MPA outcomes and, even more so, provided protection against adverse outcomes for large, mobile species of commercial (and recreational) importance. As Halpern (2014) highlighted, it is difficult to achieve all five conditions. The results here showed that isolation alone or in combination with only one or two other conditions may help achieve key MPA objectives, a positive development given recent announcements of large and isolated MPAs being approved internationally (e.g., in the Pitcairn Islands and Chile’s Nazca-Desventuradas Marine Park). The importance of ecological isolation may also help focus deliberations for future MPA site selection and suggests caution regarding opportunistic designation of MPAs based on factors related solely to managerial convenience.

As Halpern (2014) further emphasized, “more research is needed to better understand the generality of the authors’ (Edgar et al., 2014) results, and how other factors could influence conservation success” (p.168). The relatively low coverage of NEOLI pathways to positive ecological outcomes suggests that there are indeed factors other than NEOLI conditions that have an important impact on MPA success. For researchers, this points to the pressing need to further expand the scope of research on the determinants of MPA success so as to identify and account for ecological and human-oriented conditions that, in combination, help achieve recovery and protection targets across a broad range of MPA configurations.

Supplemental Information

Data S1 MPA dataset (from Edgar et al., 2014)

Raw data.

Click here for additional data file.

I thank Graham Edgar for supplying data and Mahmoud Sarhan for exploratory research during his MSc research at University of York.

Additional Information and Declarations

Competing Interests

Author Contributions

Data Availability

The authors declare there are no competing interests.

Murray A. Rudd conceived and designed the experiments, performed the experiments, analyzed the data, contributed reagents/materials/analysis tools, wrote the paper, prepared figures and/or tables, reviewed drafts of the paper.

The following information was supplied regarding data availability:

The research in this article did not generate any raw data. A summary of the existing dataset is provided as Data S1.

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
