# Peer review of "Pathways from marine protected area design and management to ecological success"

_PeerJ, doi:10.7717/peerj.1424_

## Round 0.1 · original submission · Major Revisions

· Academic Editor

Major Revisions

Please consider all the suggestions in the revised version of your manuscript.

·

Basic reporting

My major objection to the paper was the statement attributed to Halpern (2014) in the second paragraph of the introduction that “we know little about how various factors combine to affect MPA performance.” This statement overlooks the wealth of theoretical literature that uses population models to predict precisely how these various factors interact to determine biological responses. For example, the important role of ‘isolation’ (i.e., having an MPA that does not span continuous habitat) is consistent with predictions of population models that include realistic fish home range movement (e.g., Moffitt et al. 2009 Ecological Applications). Thus the paper is not really placed within the context of our full ecological understanding of MPA science. A relatively recent review and summary of this theoretical literature can be found in White et al. (2011, Frontiers in Ecology and the Environment) (apologies for citing my own paper here).

Experimental design

No comments

Validity of the findings

I am not familiar with set theoretic approaches, so I hope the editor has found another reviewer with expertise in that area to evaluate the statistics. I am taking them at face value.

Additional comments

This is an interesting paper that applies a set theory analysis to a recently-published database of marine protected area outcomes (Edgard et al. 2014, Nature). The original publication by Edgar et al. had focused on the role of five criteria (NEOLI) in determining MPA ‘success’ (measured as increases in the biomass and richness of various fish species groups). However, a weakness of their analysis (in my view) is that their analysis was inherently qualitative, designating an MPA as simply having low, medium, or high levels of the 5 NEOLI criteria. Thus I think Rudd has taken a good approach in applying an inherently qualitative approach – set theory – to investigate further. This approach allowed him to investigate which particular combinations of the NEOLI metrics tended to lead to good MPA outcomes, and what proportion of the good outcomes could be explained by the NEOLI factors. Interestingly his results cast doubt on the overwhelming importance of NEOLI, while still agreeing with Edgar et al’s finding that isolation (I) is the most consistently important of the factors.

Overall I found the paper well-written and the conclusions consistent with the data. The explanation of set theory and the various models was a steep learning curve, but with careful reading I was able to understand it. The author might consider including a sort of simple, dummy example of the set theory approach using made up data in order to explain to unfamiliar readers (i.e., the MPA community) how to interpret these results.


Minor comments:
L112: I think this definition of ‘sufficient’ should read “the outcome occurs whenever the condition is present, but also in its absence”

L189: This statement on whether Isolated is necessary seems flawed. Isolated could be necessary but not sufficient, which would fit with the pattern reported here. However it would be correct to say that “it may appear that Isolated is sufficient…but”

L409: Just a comment on Edgar et al’s claim that the increase was exponential: they found an exponential increase in the back-transformed log response ratios of each variable (i.e., inside:outside biomass). Because the metric is a ratio it will tend to produce “exponential” patterns even if the numerator and demoninator are diverging in a linear, non-exponential way. In other words they tend to amplify large results.

Reviewer 2 ·

Basic reporting

I find the paper hard to follow.

Experimental design

I could not repeat this work.

Validity of the findings

It is hard for me to validate the findings.

Additional comments

The paper uses a logic set approach to evaluating protected areas to determine how many features are associated with metrics of success. I mostly find the paper hard to follow and evaluate and imagine it needs more background in these methods to be intelligible. The authors could help by giving more details of the data and methods and working through some basic examples. I am sure I could not repeat this work but it is hard to know who is at fault here the author descriptions or my lack of knowledge. Below are some comments that arose while reading it.


L11 – The idea of resilience is important but if this is not being addressed in this paper why have it as a concluding sentence on a paragraph. It would seem more appropriate to bring up design principles.


L35 - I think more should be said about the statistical analysis used by Edgar et al in order to contrast with what is being done in this paper. I would personally like to see the data analyzed in more than one way to see if the results or conclusions differ based on the methods used, say multiple regressions versus set theory.


L47 - I think more should be said about the data in terms of how the sites were chosen, etc. The issue of selectivity is always an issue when evaluating MPAs. Even the larger question of what an MPA is important, as all it signifies is the area is protected but how and how well are not always very clear. The reader should not be sent to another paper supplement to get this information. It should be summarized so this paper can be evaluated and stand alone.

It is also not clear about what the data are in terms of their qualitative aspects and scales, etc. How might this affect the analysis and did the data need “massaging” to be useful for the set theory? All of this is critical to evaluating the paper from a reviewer’s perspective.

L53 – This section is also too brief to really evaluate the methods if one has not read the papers or are not familiar with the software. Given that this is not standard practice, the methods are insufficient to evaluate. How are crisp and fuzzy differentiated? Can low, medium, and high be better described? Is this relative this this data set or to some objective metrics, for example? Can more be said about how Edgar differentiated the MPAs? There are objective metrics for biomass, such as MacNeil et al. 2015.


L74 How can one evaluate biomass OR diversity. These are two separate units so they each must be evaluated separately. So, here is it clear that the evaluation is relative to the whole data set. This is a problem because most MPA studies either have to compare to a control site or they have to have some objective values, such as a pristine baseline or something. This means one is only comparing MPAs to other MPAs but not the social or environmental context of the MPA relative to control sites.


L82 – No MPAs are close to pristine in most of the studies that I have seen, so I am not sure why this statement would be made here except to cover the problem above? The pristine issue is quite complicated and so this needs more effort to make it clear what this means and what the actual values are and why the cutoffs were used. Otherwise, it just seems quite arbitrary or relative where success is relative to other MPAs but of which many are in different in ways that are not well described or understood.

As an example, units in table 2 make little sense to me unless these are somehow log values. One should be able to undesrstand the actual units. For example, the high biomass given in table 2 is 12 g/250 m2 if I understand the units, which equals 0.480 kg/ha. This is a very small number and hard to compare with say what MacNeil 2015 calls Pristine biomass or 1000 kg/ha. I assume the 12 is not grams per 250 m2 but what is it? See what values are in remote wilderness areas for comparison – ie. Sandin, Friedlander, Graham. This kind of issue of lack of unit transparency can greatly influence the ability to evaluate the paper and the author’s interpretations of success. So, this needs to be more explicit and the data need to be very clear in terms of what units are, etc.

How are the sties in table 3 organized?


I am finding the logic sections and most of the results hard to follow and they might be less abstract and more specific to this case study. Perhaps, this needs another reviewer more familiar with logic sets.

---

## Round 0.2 · Minor Revisions

· Academic Editor

Minor Revisions

Please consider the remaining suggestions in a revised version.

·

Basic reporting

No comments.

Experimental design

No comments.

Validity of the findings

No comments.

Additional comments

Overall I think Rudd has done a good job of responding to the reviewers’ comments. I still had two outstanding concerns:

1) I am still confused by the text regarding whether Isolated is necessary or not (L285ff in the revised ms). I agree with the statement in the rebuttal that Isolated has already been tested for being necessary and is not. However this statement:

“While it may appear that Isolated is necessary for High biomass outcomes, recall that the definition of a necessary condition is that it is a superset of the outcome: the outcome only appears in the presence of the condition. Table 6 showed, however, that row 12 (neOlI) had three empirical instances where High biomass was not achieved even though the MPAs were isolated; if Isolated were a necessary condition, these MPAs would also have exhibited a High biomass outcome.”

Does not make sense to me. I agree with the definition of “necessary”. However the evidence that would show Isolated is not necessary would be empirical instances in which the MPA had High Biomass even though it was not Isolated. The reported pattern, that High Biomass was not achieved even though the MPAs were Isolated, is not inconsistent with being necessary (Isolated could be necessary, but other factors are also required to obtain High Biomass). However, the empirical pattern *is* inconsistent with Isolated being Sufficient: if there are Isolated MPAs that do not have High Biomass, then isolation is not sufficient to generate high biomass. This is why in my original comment I suggested that this passage should be about Isolated’s sufficiency, rather than its necessity. The revised passage still seems to have the same problem, unless I am completely missing something here.

2) Regarding the claim of Edgar et al. that increases were exponential: the response letter notes that a change was made and a note about the problem of back-transformed log ratios was inserted into the text. However I do not see this new text in the revision in the relevant section (around line 532); what line is it on?

Finally, I agree with Rudd’s responses to the second reviewer; because this dataset is already published by Edgar et al., there is no need to rehash all of the details of data collection or the original analysis.

---

## Round 0.3 · accepted · Accept

· Academic Editor

Accept

Thank you for giving us the opportunity to publish your results.